# Assessment of Hemp Seed Oil Quality Pressed from Fresh and Stored Seeds of *Henola* Cultivar Using Differential Scanning Calorimetry

**DOI:** 10.3390/foods12010135

**Published:** 2022-12-27

**Authors:** Mahbuba Islam, Yolanda Victoria Rajagukguk, Aleksander Siger, Jolanta Tomaszewska-Gras

**Affiliations:** 1Department of Food Safety and Quality Management, Poznań University of Life Sciences, ul. Wojska Polskiego 31/33, 60-637 Poznan, Poland; 2Department of Biochemistry and Food Analysis, Faculty of Food Science and Nutrition, Poznan University of Life Sciences, Wojska Polskiego 31, 60-634 Poznan, Poland

**Keywords:** cold-pressed hemp seed oil, differential scanning calorimetry, thermo-oxidative stability, isothermal test, non-isothermal test

## Abstract

Cold-pressed hemp (*Cannabis Sativa* L.) seed oil has become very popular amongst consumers and researchers, due to its manifold application in food and medicine industry. In this study, oils pressed from stored and fresh hemp seeds of the *Henola* cultivar were analyzed. Determination of the acid value (AV) and color of oil (a* parameter) revealed significant differences between the two groups of oils (fresh and stored seeds) in contrast to the peroxide value (PV), p-anisidine value (p-AV), and fatty acid composition. On the other hand, isothermal and non-isothermal assessments of the thermo-oxidative stability by differential scanning calorimetry (DSC) showed no significant differences in oxidation induction time (OIT) as well as in onset temperature (T_on_) between two groups of oils (*p* > 0.05). The DSC isothermal test (OIT 160) showed significant correlations with mono- and polyunsaturated fatty acids as well as with values of AV and a* (*p* ≤ 0.05), in contrast to the non-isothermal test, for which correlations were not significant (*p* > 0.05). However, the best distinction of both groups of oils was obtained analyzing all results together (DSC, fatty acid and tocochromanols composition, color, and oxidative stability results) by principal component analysis (PCA).

## 1. Introduction

By dint of being highly popular amongst consumers and scientists, there has always been a research interest in hemp seed oils among researchers all over the world. This plant-based non-traditional oil has now been recognized as an emerging source of nutrient-rich nutraceuticals. Commonly known as industrial hemp grown for its fibers, the variety of hemp (*Cannabis Sativa* L.; *Cannabaceae*) is also a nutritional crop, which has been cultivated for centuries to produce a diverse range of useful products; i.e., seeds; oils; fiber; medicine; food supplements; cosmetic; narcotics; and most recently and promisingly, insulation, and bio-composite materials as biodiesel, which meets the global demand for sustainable energy [1,2,3,4]. This annual plant has been used and later cultivated by humans since pre-historic times [5,6]. Originating in western and central Asia, this cosmopolitan species has been introduced and cultivated all over the geographical zones for centuries due to its adaptability towards climate changes [7]. Upholding the century-old tradition, European countries like France, the Netherlands, Germany, Great Britain, Spain, Poland, Italy, and Austria have all produced hemp as an industrial crop since the Middle ages [3]. In 2018, the European Industrial Hemp Association (EIHA) reported that since the legalization of industrial hemp cultivation in 1993–1996, the surface area of hemp culture in European states has increased by 614% yearly, on average, with a record high 50,081 ha area dedicated to cultivation [2,8]. Keeping up with subsidized EU hemp production, in 2017 Poland registered its highest surface area of hemp cultivation (7000 ha) recorded in the last 30 years, as reported by the Institute of Natural Fibers and Medicinal Plants INF&MP [3]. Until 2021, 11 varieties of industrial hemp crop had been listed and reported by the Central Research Center for Varieties of Cultivated Plants (COBORU) in Poland, amongst which one is a dioecious variety—*Matrix*, while the rest are monoecious i.e., *Beniko*, *Bialobrzeskie*, *Glyana*, *Henola*, *Mietko*, *Rajan*, *Sofia*, *Tigra*, *Wielkopolskie*, and *Wojko* [2,9]. Therefore, the cultivation goal has always been predominantly oriented with long bast fiber and straw production; harvested seeds (commonly as a by-product) from the cultivation regime are heterogenous in quality. Hemp seed oils are most commonly produced as cold-pressed oils (often referred as virgin hemp seed oil), which has a distinctive nutty taste and a light to dark-green color. Researchers have shown that the oil yield from hemp seeds ranges between 25 and 35% [6]. Numerous factors may influence the oil quality in general, i.e., the geographical location of production, types of cultivars and seeds, cultivation conditions and genetic predisposition of varieties, extraction procedure, and storage condition before reaching consumers [10,11]. To access oil quality beforehand, assessing oxidative stability is the major parameter. Therefore, studies report that thermal degradation and oxidation of cold-pressed oils depends not only on fatty acid composition but also on the antioxidant compounds and other lipid compounds [12]. Since cold-pressing does not require chemicals or organic solvents and is done by means of a screw/mechanical press or a hydraulic press [13], it has been reported that numerous bioactive compounds along with a wide array of essential amino acids are retained in the final product without any chemical contamination [14]. Experiments published to date have proved hemp seed oils to be an exceptionally rich source of unsaturated fatty acid content, i.e., up to 80–90% of PUFA, 1–5% of γ-linolenic acid (GLA), and a unique presence of stearidonic acid (SDA) 0.5–2%. The optimum ratio of two essential fatty acids (EFA’s) linoleic acid: α-linolenic acid (LA:ALA) as 3:1 has been studied, as this ratio is the most desirable ratio of n-6 and n-3 for human consumption [7,15]. In addition, GLA and SDA act as biological precursors for longer chain omega-3 fatty acids, i.e., eicosapentaenoic acid (EPA; C20:5n-3) and docosahexaenoic acids (DHA; C22:6n-3) rapid synthesis in the human body [6,16]. The presence of significant quantities of antioxidative compounds like tocochromanols or carotenoids has been reported in hemp seed oil [17]. Another significant compound found in cold-pressed hemp seed oil is chlorophyll, which unlike tocochromanols, can act as a prooxidant and increase the susceptibility of photo-oxidation and changes the color of the oil from dark green to yellow [14].

Maintaining the oxidative stability of hemp oil at a constant level is a prerequisite for obtaining the desired sensory quality. However, it depends not only on the compounds present in it, but also on the quality of the seeds. Apart from pressing them for oil, hemp-seed foods are popular throughout the Baltic states, North America and Europe, middle East, and China. As a result, storing the seeds for various purposes throughout seasons is a common phenomenon [7]. However, the unique composition of hemp seeds and oils pressed from them has been highlighted in literatures with the abundance of chlorophyll and PUFA content, which results in a rapid complex series of chemical reactions that lead to deterioration of the seeds and oils [18]. Thermal characteristics of hemp seed oils analyzed using thermogravimetric analysis (TGA) [19], or pressure differential scanning calorimeter (PDSC) [20] is able to provide copper-bottomed database to assess the oxidative stability parameters in isothermal and non-isothermal conditions. Additionally, the comprehensive analysis of such vegetable oils using DSC was provided with significant results to characterize oil quality in terms of stability [12]. Only a few studies have been published regarding the oxidative stability of hemp seed oil [1,19], while more specifically, to the best of the authors’ knowledge, to date none have been published on the Polish hemp cultivar *Henola*, which was registered in the Central Research Center (COBORU) list in 2017 [9]. Hence, the aim of this research is to compare two groups of cold-pressed oils from the hemp seed cultivar *Henola* (fresh and 1-year-old seeds) in terms of their thermal and chemical oxidative stability properties using differential scanning calorimetry and conventional chemical methods.

## 2. Materials and Methods

### 2.1. Materials

Cold-pressed hemp seed oils originate from seeds cultivated from the *Henola* cultivar in the Greater Poland region. For this research, samples were purchased in two seasons, 2018 and 2019, from different suppliers. Seeds harvested at the end of November 2018 are considered as group 1 and named HL_18A and HL_18B, while seeds harvested at the end of November 2019 are group 2, and are marked as HL_19A, HL_19B, and HL_19C. At the beginning of February 2020, all the seeds were cold-pressed by maintaining a temperature below 50 °C and then left for the 24-h decantation process. Following this, the oils were restored in the laboratory in brown glass bottles, and storage conditions were maintained at sub-zero temperatures (−80 °C).

### 2.2. Methods

#### 2.2.1. Fatty Acid Composition

To determine the fatty acid composition, all the samples were analyzed in two replications. Gas chromatography-flame ionization detector (GC-FID) was employed for these experiments. Around 15 mg of oil was dissolved in 1 mL of hexane (for HPLC, Sigma Aldrich, St. Louis, MO, USA). After adding 1 mL of 0.4 N sodium methoxide, 5 mL of distilled water was added after 15 min and the top layer was separated for analysis. According to the official AOCS method [21], Trace 1300 chromatograph (Thermo Fisher Scientific, Waltham, MA, USA) were used to analyze fatty acid methyl esters. Supelcowax 10 capillary column (30 m × 0.2 mm × 0.2 μm) was employed to perform separation. An injection was performed in splitless mode. Sample volume was 1 µL. During the experiment, hydrogen gas was used as carrier gas while the initial furnace temperature was set to 160 °C and was increased from 12 °C min^−1^ to 220 °C. A final temperature of 220 °C was kept for 20 min. The identification of fatty acid methyl esters was done by comparing the retention times in the sample and in the 37-Component FAME Mix (Supelco, CRM47885, Sigma-Aldrich, St. Louis, MO, USA).

#### 2.2.2. Color Measurement

To determine the color of the oil samples, Konica Minolta CM-5 spectrophotometer and SpectraMagicNx software were used. After calibrating the transmission chamber using CM-A213 zero calibration plate (black calibration), followed by distilled water in a 10-mm CM-A98 glass cuvette (white calibration), the instrument was ready to measure both translucent and transparent liquid samples. The study was conducted using the Hunter Lab scale. In Hunter Lab scale assessment, parameter L*, a*, and b* were measured, where L* denoted the lightness of the color black to white (range from 0 to 100), parameter a* was determined by green (when the value is below 0) and red (when the value is above 0) tinge, and the b* parameter represents the color blue (when value is below 0) and yellow (when value is above 0). The color measurement was performed in three replications.

#### 2.2.3. Determination of Oxidative Stability by DSC

Oxidative stability of the samples was determined by following ISO 11357-1 [22] and also implementing the ASTM D3895-14 [23]. DSC 7 Perkin Elmer (Waltham, MA, USA) along with an Intracooler II was employed for the experiment, which was operated with Pyris software 13.3. The oxidative stability characteristics determination was done by following both the isothermal and non-isothermal protocol. The instrument was calibrated using indium (m.p. 156.6 °C, ΔH_f_ = 28.45 J/g) and n-dodecane (m.p. −9.65 °C, ΔH_f_ = 216.73 J/g). During the experiment, 99.99% pure nitrogen gas was used as the purge gas. Approximately 6–7 mg of oil the sample was weighed into 50 µL open aluminum pans (Perkin Elmer, No. 18 02190041). The pan with the sample was weighed as well as an empty pan (as reference pan), then placed in the instrument’s chamber. For the isothermal experiment, data was acquired for the targeted temperature programs of 120 and 140 °C. A constant oxygen flow of 20 mL min^−1^ (purity 99.995%) was maintained throughout the measurement time. The parameters used for the assessment if the thermal phenomena were oxidation induction time (OIT), oxidation end time (OET), length of oxidation Δt = OET − OIT, and rate of oxidation, determined based on acquired curves. To determine OIT value, the DSC oxidation curve was normalized, based on the intersection of the extrapolated baseline and the tangent line to the descending exotherm. On the other hand, the end of the propagation and start of the termination stage of oxidation were calculated to determine the OET value at the minimum level of the heat flow of the exotherm. The oxidation rate was calculated according to the following equation:(1)Oxidation rate=(Y1−Y2)/Δt 
where Y1 represents heat flow at OIT point [W/g], Y2 represents heat flow at OET [W/g], and Δt the length of oxidation [min].

The non-isothermal analysis was carried out for two different heating rates i.e., 2 and 5 °C/min. The oxygen flow was carried out at 20 mL/min. From the DSC curves obtained during the experiment, the onset temperature (T_on_) and the end temperature (T_end_) were calculated. Onset temperature was determined as the intersection of the extrapolated baseline and the tangent line to the descending curve of the recorded exotherm and denoted as Ton, while Tend was measured as the temperature at the minimum value of the heat flow, which is an expression to the end of the propagation and beginning of the termination stage. For all DSC experiments, samples were analyzed in two replications.

#### 2.2.4. Chemical Determination of Oxidative Stability

The p-anisidine value (p-AV) measurement was performed to assess the level of secondary oxidative products (i.e., aldehydes, carbonyls, trienes, ketons) present in the samples according to ISO 6885:2016 [24]. Spectrophotometric measurements were performed with a quartz cuvette with a 10-mm optical path length. Peroxide value (PV) was determined to measure the peroxide present in the sample in order to indicate the primary oxidation of the oils by following the ISO 3960:2007 procedure [25]. The total oxidation value (TOTOX) parameter was calculated based on the pAV and PV values to determine the overall oxidation profile of the sample by means of the calculation based on the formula TOTOX = pAV + 2PV. As an indicator of the degree of hydrolytic changes, measurement of acid value (AV) can express the oxidative condition of the samples as free fatty acids, which are caused by the decomposition of triglycerides. AV was measured according to the official AOCS method [26]. Conjugated dienoic acid and trienoic acid value determination were carried out using Cary 1E spectrophotometer by following the Ti 1a-64 method [27]. Spectroscopic measurement at specific extinction (K_233_, K_268_) was measured at wavelengths λ = 233 and λ = 268, by following the AOCS standard method [28]. All chemical analyses of the oil samples were done in three replications.

#### 2.2.5. Radical Scavenging Activity by DPPH (RSA DPPH)

The antioxidant activity of the oil samples was measured using DPPH radicals (2,2-diphenyl-1-picrylhydrazyl) [29,30]. Firstly, 100 µL of a methanolic solution containing a sample extract or standard solution (Trolox) was mixed with a methanolic solution of DPPH radicals (1 mM, 250 µL) and 2 mL of methanol 80%. The mixtures were thoroughly vortexed and left in the dark. After 20 min, the spectrophotometric measurements were performed 10 times with a Varian Cary 1E (Berlose, Australia) using ethyl acetate as a blanc solution. Antioxidant analysis was performed with three repetitions.

#### 2.2.6. Determination of Tocopherols and Plastochromanol-8 by NP-HPLC

Initially, the oils sample (200 mg) was dissolved in n-hexane, resulting in a 10 mL solution, and transferred to vials for the analysis of tocopherols and plastochromanol-8. Instruments employed to qualitatively and quantitatively identify the tocopherols were performed by using a Waters HPLC system (Waters, Milford, MA, USA) consisting of a pump (Waters 600), a fluorimetric detector (Waters 474), a photodiode array detector (Waters 2998 PDA), an autosampler (Waters 2707), a column oven (Waters Jetstream 2 Plus), and a LiChrosorb Si 60 column (250 × 4.6 mm, 5 µm) produced by Merck (Darmstadt, Germany). A mixture of n-hexane with 1,4-dioxane (96:4 *v*/*v*) was used as a mobile phase with a flow rate of 1.0 mL/min in the method. During the assessment of the fluorescence of tocopherols and PC-8, the excitation wavelength was set at λ = 295 nm and the emission wavelength at λ = 330 nm. Standards of α-, β-, γ-, and δ-tocopherols (>95% of purity) were purchased from Merck (Darmstadt, Germany). The plastochromanol-8 contents were assayed and calculated according to the procedure outlined by [31].

#### 2.2.7. Statistical Analysis of Results

The results obtained from both traditional chemical and analytical experiments were presented in the form of mean and standard deviation. Hartley–Cochran–Bartlett test was adopted to verify the variance homogeneity. Later, for variance homogeneity, one-way analysis of variance (ANOVA) was used and Tukey’s test was applied to create statistically homogeneous groups. However, if the variances were not homogeneous, non-parametric tests, i.e., ANOVA and the Kruskal–Wallis rank test were performed. Principal component analysis (PCA) was performed to illustrate the relationships between variables. The PCA test was also able to detect some important patterns between variables and objects. As a result, it was possible to present the dataset as reduced from a higher to a lower dimensional level. Statistical analysis of the results recorded was performed using Statistica 13.3 software (TIBCO Software Inc., Palo Alto, CA, USA) at a significance level of α = 0.05.

## 3. Results

### 3.1. Physicochemical Characteristics of Hemp Seed Oil

#### 3.1.1. Color Measurement

Color measurement of hemp seed oil samples was carried out using a CIELAB color system. Figure 1 shows the samples of hemps seeds and oils pressed from the seeds.

Parameters of L*, a*, and b*, presented in Table 1 show significant differences amongst the oil samples (*p* ≤ 0.05). The lightness of the oils expressed by the L* value ranges between 33.00 and 53.46, and the b* value represents the yellowness of the oils, which ranged between 56.48 and 88.20. Interestingly, in group 1 there were both higher and lower values of L* and b*, compared to group 2, which indicates that the storage of seeds does not affect these parameters. In the case of a*, significant differences between the two groups of oils were visible (*p* ≤ 0.05). The study shows that the lower values of a* are connected with the greenish color, which might result from chlorophyll pigments [32]. As for samples HL_18A and HL_18B, the highest a* values were observed, 10.06 and 11.81, respectively, whilst for group 2 these were from 5.68 to 7.73. Despite these values being slightly higher than the results obtained by other authors [33], the differences in the a* parameter between the two groups suggests the influence of seeds storage and the lowering of the greenness in group 1 can be explained by the chlorophyll oxidation.

#### 3.1.2. Fatty Acid Content

Table 2 projects how the fatty acid content differs between hemp seed oil samples. The fatty acid composition of hemp seed oils shows the predominance of polyunsaturated fatty acids (PUFA) amongst unsaturated fatty acids, ranging from 74 to 76%. The highest PUFA content was achieved for sample HL_18B from group 1, which was significantly different from all samples (*p* ≤ 0.05). In turn, for the sample HL_19A from group 2, the highest value of monounsaturated (MUFA) was noted (14.59%), which was significantly different from all samples of group 1 (*p* ≤ 0.05). The findings about the fatty acid composition of *Henola* variety are comparable to the results published by other authors for the same cultivar [2] and for another cultivar, i.e., *Fassamo* cultivar [1], even oils extracted from roasted hemp seeds showed a similar range of PUFA content, 75.8–77.2% [34]. The total degree of unsaturation was 88.73–89.15% for the *Henola* cultivar, which is similar to the authors’ experiments on other Polish hemp seed oils from the local market [35]. An abundance of essential fatty acids (EFA) was detected for the oils obtained from the *Henola* cultivar as 72–74% for all varieties, and this was similar to the results obtained for oils extracted by Soxhlet extraction (74–77%) [36]. Saturated fatty acid (SFA) content ranged from 10 to 11%, whilst a ratio of UFA to SFA was from 8.04 to 8.44. In this study, the ratio for n-6/n-3 fatty acids was established as 1.90 to 3.09, which is comparable to other authors’ findings [34,36].

#### 3.1.3. Tocochromanols Content and DPPH Radical Scavenging Activity

Tocochromanols content was quantified for all the cold-pressed samples of the *Henola* cultivar, which are present as natural antioxidants as shown in Table 3. Four different derivatives of tocopherols are found in nature, i.e., α-, β-, γ-, and δ-, based on the methylation of the chroman ring. Another compound is defined as plastochromanol-8, which is structurally more similar to that of γ-tocopherol. By controlling the radical chain reaction in PUFAs, tocopherols are known to be responsible for preventing the oxidation of oils during production and storage. As can be seen from Table 3, the most abundant is γ-tocopherol, ranging from 63.47 to 73.78 mg/100 g, which has been reported to be a peerless quality for hemp seed oils compared to the other vegetable oils [7]. The next abundant example is α- tocopherol (4.36 to 6.19 mg/100 g), which is known to have the greatest antioxidative properties. Trace amounts of β-tocopherol (0.11–0.15 mg/100 g, absent in HL_18B), and δ-tocopherol (1.14–1.47 mg/100 g), are also present in the hemp seed oils. Overall, total phenolic content (TPC) ranged from 69.11 to 81.43 mg/100 g and was significantly different for all samples (*p* ≤ 0.05). Quantitative analysis of tocopherols from this study is comparable to the study on hexane-extracted hemp seed oil conducted by [1], and of enzymatically-treated hemp seed oils [37]. Generally, for α- and γ-tocopherol and TPC, significantly higher values were detected in group 1 oils than in group 2 (*p* ≤ 0.05). Plastochromanol-8 (PC-8) was also detected, as presented in Table 3, and it ranged from 0.26 to 1.64 mg/100 g; in sample HL_18A it was absent. PC-8 is structurally similar to γ- tocopherol (methyl groups reside in positions 2, 7, and 8 in the chromanol-6 ring) and is reported to have a stronger antioxidant profile than α-tocopherol [38]. The presence of this compound in hemp seed oil has also been reported by other authors [14].

DPPH radical scavenging activity of hemp seed oils, expressed as % of inhibition, was also measured (Table 3). Ranging from 6.96 to 9.11 µmol TE/g oil, DPPH value determination of these cold-pressed hemp seed oil samples shows the potential to minimize the oxidation damage mediated by free radicals. Compared to the data in the literature, it is comparatively lower than, for example, hexane-extracted nutmeg seed oil (16.46 µmol TE/g oil), but higher than white mustard, anise, coriander, and caraway seed oils (4.21, 3.44, 2.24, and 6.32 µmol TE/g oil, respectively) [39]. In the data presented in Table 3, it can be seen that for group 1 of hemp seed oils the values of DPPH scavenging activity were higher than for group 2. DPPH values also show that comparatively strong antioxidant activity is exhibited by group 1 oils, and 8.53 and 9.11 µmol TE/g oil for HL_18A and HL_18B, respectively, (not significantly different, *p* > 0.05).

#### 3.1.4. Oxidative Stability Determination by Chemical Methods

Table 4 displays the results of oxidative stability assessment, done by different chemical analytical methods, such as acid value (AV), peroxide value (PV), p-anisidine value (p-AV), dienoic and trienoic acid value; in addition, specific extinctions (K_233_, K_268_) are presented. PV values were measured in the range 11.01–18.80 (meq O_2_/kg), and the values were significantly different for each sample (*p* ≤ 0.05). These values are comparable to the values obtained for commercial hemp seed oils (4.31–22.14 meq O_2_/kg) [32]. AV values show significant differences between two groups of oils, as they were significantly higher for group 1 (18.53 and 21.16 mg KOH/g) than for group 2 (10.30 to 10.33 mg KOH/g) (*p* ≤ 0.05). This observation confirms the well-known fact that the storage of seeds caused the hydrolysis of triacylglycerols and the release of free fatty acid, a phenomena that can be measured by the increase of AV value. The obtained values are two to three times higher than the values obtained for commercial hemp seed oils [40]. Secondary oxidation products generated in the oils were also measured by determining the p-anisidine value (p-AV), which ranged from 0.97 to 1.34, while no significant differences were stated between the oil samples (*p* > 0.05). Presented data are similar to the results obtained by different authors [32,41]. The overall oxidation state of the oils was calculated and expressed as the TOTOX value, which ranged from 23.36 to 38.70, and showed a significant difference amongst all samples (*p* ≤ 0.05). Meanwhile, considering the TOTOX value, HL_19C was characterized as the most stable oil with the lowest value of 23.36. Furthermore, in order to reveal the status of oxidative deterioration of the oils, conjugated dienoic acid (%) and trienoic acid (%) values were determined (Table 4). During the oxidation of unsaturated fatty acids, conjugated hydroperoxides were formed due to double bonds changing to a single bond, and the content of these compounds can be expressed as the percentage of dienoic acid value (if two double bonds C=C are altered, it is the primary oxidation state) and also trienoic acid value (if three double bonds C=C are altered, it is the secondary oxidation state). For the hemp seed oil samples, dienoic acid values were measured between 0.1 and 0.22, while the trienoic values ranged from 0 to 0.16. In addition, the specific extinction coefficients of ultraviolet absorption (K_233_ and K_268_) measured at wavelengths λ = 233 and λ = 268 are also presented, which are capable of outlining the primary (i.e., peroxides, dienes, and free fatty acids) and secondary oxidation products (i.e., aldehydes, ketones, and trienes) present in oil samples. As can be seen in Table 4, the values of K_233_ ranged between 0.15 and 0.30, and K_268_ values are between 0.03 and 0.07. These results are comparatively low compared to the results obtained by other authors [1,33,37].

### 3.2. Oxidative Stability Assessment Using Differential Scanning Calorimetry (DSC)

#### 3.2.1. Isothermal DSC Test

Differential Scanning Calorimetry (DSC) was employed to determine the thermo-oxidative stability of two groups of cold-pressed hemp seed oils from *Henola* cultivar. Primarily, isothermal assessment was carried out at a 120, 140, and 160 °C temperature program (Table 5). Curves obtained from experiments reveal the oxidation process as an exothermic reaction between the sample and the oxygen flow at specific temperatures (Figure 2). Treating hemp seed oils in an oxygen atmosphere with constant exposure to a high temperature initiates the formation of hydroperoxides from the UFAs, which in turn leads to thermal degradation accompanied with off-flavor compounds. Since the oxidation process at room temperature is a relatively slow phenomenon, the idea of accelerated shelf life testing (ASLT) by elevation of factors like light or temperature is commonly practiced by researchers [42]. Advancing this concept, the application of DSC allows for a given temperature measurement of oxidation induction time (OIT) as the intersection of tangent straights of the DSC curves. In Figure 2, isothermal curves obtained for a 120 and 140 °C temperature program are presented, and in Table 4, the OIT parameters calculated for 120, 140, and 160 °C isothermal conditions are given.

Oxidation induction time (OIT) indicates the initiation of oxidation, which is the reaction stage between the radicals, formed during the induction time and PUFA as shown in Equation (2). A decrease in the heat flow is related to the propagation stage, where oxygen is consumed in the reaction according to Equation (3). Oxidation end time (OET) expresses the termination stage, where stable products are formed (Equation (4)).
Initiation               R-H + R˙ → R˙ + H˙(2)
Propagation               R˙ + O_2_ → ROO˙                             ROO˙ + RH → ROOH + R˙(3)
     Termination (stable products)     ROO˙ + ROO˙ → ROOR + O_2_                    RO˙ + R˙ → ROR                   R˙ + R˙ → RR(4)

For all oil samples, the OIT values were shorter with the increase of the isothermal temperature. The parameter of OIT determined at 120 and 160 °C showed significant differences between hemp oil samples, regardless of whether they belonged to the first or the second group (*p* ≤ 0.05). In the case of OIT at 140 °C, there were no significant differences between oil samples except the sample HL_19C from group 2, for which for all temperatures (120, 140, and 160 °C), the highest oxidation resistance was noted, as the OIT values were 58.18, 14.99 and 1.13 min, respectively. Comparing both groups, it can be noticed that both in group 1 and group 2 there were high and low values, which proves that the storage time of seeds did not affect the OIT values (Table 4). Analyzing oxidation end time (OET), expressing the end of the propagation process of oxidation, analogous thermal behavior for all samples can be observed, as it was for OIT. It can be seen that the values obtained are significantly different (*p* ≤ 0.05) among all hemp oil samples, and for sample HL_19C, the highest time to complete the first stage of the oxidation process for all three isothermal programs was detected i.e., 80.9, 27.4, and 9.4 min, respectively. The findings from this analysis are unique in such a way that instead of adopting the DSC instrument, other authors reported the stability profile of hemp seed oils using TGA [19] or PDSC instruments [20]. A similar possible comparison from TGA analysis shows that the isothermal program at 70, 80, and 90 °C initiates the oxidation process for cold-pressed hemp seed oils at 368.71 min, 198.06 min, and 96.31 min, respectively [19].

#### 3.2.2. Non-Isothermal DSC Test

DSC non-isothermal assessment provided data on the oxidation temperature of hemp seed oils at different heating rates i.e., 2 and 5 °C/min (Table 6). The parameter calculated from the DSC curves is oxidation onset temperature (T_on_), which signifies the starting temperature of the oxidation process, and end temperature (T_end_), which reveals the end of the propagation process of oxidation. With an increase in the heating rate from 2 to 5 °C/min, all hemp oil samples exhibited a similar degree of T_on_ and T_end_ increase, as they were always higher for 5 °C/min than for 2 °C/min. From Table 6, it can be seen that there were significant differences between oils samples in T_on_ and T_end_ values (*p* ≤ 0.05), but they were not significantly different between two groups of oils, i.e., group 1 and 2. However, similarly as in the case of OIT, sample HL_19C from group 2 presented the highest T_on_, i.e., 148.14 °C and 163.14 °C for 2 and 5 °C/min scanning rate, respectively. The reciprocity of increasing oxidation resistance with higher scanning rates for oil samples during non-isothermal assessments is comparable with data obtained for other cold-pressed oils like flaxseed oil [24] and camelina oil [12].

## 4. Discussion

By analyzing the DSC isothermal (OIT) and non-isothermal (T_on_) parameters, it was possible to assess the proneness of *Henola* cultivar hemp seed oils to oxidation and compare it with other popular vegetable oils. From the results obtained, it can be stated that the isothermal OIT test as well as T_on_ parameters are good indicators of resistance to thermal oxidation. The assessment of thermo-oxidative stability of hemp oil in three different temperatures 120, 140, and 160 °C enables to analyze the relationship between isothermal temperature and time of oxidation for various DSC parameters (OIT, OET, ∆t, and rate of oxidation), which was expressed by the exponential function equation with high coefficients of determination (0.98, 0.88, 0.95, and 0.98, respectively), as is shown in Figure 3. The trendlines expressed by exponential function might help in predicting the initiation of oxidation for different thermal conditions.

On the other hand, other chemical compounds may also be important for the stability of hemp seed oil. Data presented in Table 4 show the samples’ stability determined using different conventional chemical methods. Peroxide values represent the presence of peroxides and hydroperoxides (known as primary oxidation products), whence p-anisidine values shows the concentration of unsaturated aldehydes (detection at 350 nm absorption), which can express the secondary oxidation metabolites present in samples. Considering the primary oxidation condition, samples shows significant differences (*p* ≤ 0.05) regardless of the freshness of the seeds, whence for secondary oxidation phenomena, the characteristics are opposite, thus pronouncing the overall freshness of the oil samples. Hence, oil sample HP_19C showed the highest stability simultaneously throughout all chemical measurements, which in turn has been supported by the results obtained from isothermal (OIT) and dynamic (T_on_) parameters from DSC. A Different approach presented shows the simultaneous increase in primary and secondary oxidation products after roasting them in different temperatures [7] or during accelerated oxidation [18]. Alternative chemical analysis to determine oxidative stability paved the results of conjugated dienoic and trienoic acid (%) values, and the ultraviolet absorption measured at wavelengths λ = 233 (K_233_) and λ = 268 (K_268_) presented for oil samples shows the primary and secondary oxidation products for both groups of oils. Although thIS data shows the notifiable presence of primary deterioration for all samples, it did not show any significant differences (*p* > 0.05) between the seeds quality of stored and fresh-pressed oils. In order to establish which parameters are significantly correlated with DSC parameters, correlation analysis was carried out (Table 7). Significant positive correlation coefficients were obtained between DSC oxidation parameters and MUFA content (at a significant level of α = 0.05). The next significant correlation (negative) was found between OIT parameters and PUFA content. Strong, significant negative correlations were also found for AV with OIT and OET values for 160 °C (r = −0.921 and r = −0.724), as well as with color parameter a* for 160 °C (r = −0.959 and r = −0.837).

Since resistance to oxidation was established for two groups of oils at three different temperatures (120, 140, and 160 °C), principal component analysis (PCA) was performed separately for each temperature to identify the relationships between the values obtained by OIT DSC and the rest of the parameters tested (fatty acid content as total SFA, MUFA, PUFA, total phenolic content, DPPH radical scavenging activity, AV, L*, a*, and b*). Figure 4 presents the loading plots for PC1 and PC2 components for physicochemical characteristics and DSC parameters determined at temperatures 120 °C (Figure 4A), 140 °C (Figure 4C), and 160 °C (Figure 4E), respectively. For all three of these, the first six principal components can explain over 99% of the total variance, from which more than 70% can be explained by the first two factors: PC1 and PC2. From Figure 4A, it can be seen that the most positive correlation with PC1 is shown by AV (0.86), PUFA (0.91), total phenolic compound (0.71), and a* (0.67). Meanwhile, the highest negative correlation with PC1 was demonstrated by MUFA (−0.93). Analyzing the next figures, (i.e., Figure 4C,E), data shows that the strongest negative correlation with PC1 was projected by MUFA content (−0.98 for Figure 4C and −0.95 for Figure 4E), whilst the strongest positive correlation was for PUFA content (0.96 for Figure 4C and 0.92 for Figure 4E, respectively). From the perspective of DSC parameters, OIT values for 120, 140, and 160 °C show a weak negative correlation with PC1 (−0.37 for 120 °C, Figure 4A; −0.45 for 140 °C, Figure 4C and −0.47 for Figure 4E). Figure 4 also shows the distribution of all *Henola* hemp seed oil samples in three graphs for each temperature program separately, for 120 °C (Figure 4B), 140 °C (Figure 4D), and 160 °C (Figure 4F), where the projection of cases shows the distinction of group 1 oils from group 2. Analyzing the graphs, it can be seen that for the 120, 140, and 160 °C temperature program, group 1 of oils pressed from stored seeds was always located on the positive side of PC1. The graphs showing the distribution of hemp seed oils samples clearly demonstrate that all three isothermal temperature programs considered with other chemical parameters are able to differentiate between groups of oils based on their seed’s freshness.

## 5. Conclusions

The significance of research related to oxidation phenomena in food products is obvious in terms of its nutritional value and safety. First and foremost, the cumulative popularity of hemp seed oils for their many applications in different industries requires the characterization of this oil at an advanced level. Differential scanning calorimetry (DSC) was used successfully in this study to characterize the thermal behavior of cold-pressed hemp seed oil under isothermal and non-isothermal conditions. Comprehensive physicochemical analyses of hemp oils were also carried out and they showed that for the acid value (AV), the color of oil (a* parameter) and tocochromanols content were significantly different between the two groups of oils (fresh and stored seeds) (*p* ≤ 0.05), in contrast to the peroxide value (PV), p-anisidine value (p-AV), and fatty acid composition. DSC isothermal and non-isothermal assessments of the thermo-oxidative stability also showed no significant differences in oxidation induction time (OIT) as well as in onset temperature (T_on_) between two groups of oils (*p* > 0.05), except the OIT values determined for temperature of 160 °C, for which significant correlations with mono- and polyunsaturated fatty acids as well as with values of AV and a* were stated (*p* ≤ 0.05). In turn, for the non-isothermal DSC test, all correlations analyzed were not significant (*p* > 0.05). Based on the results obtained, it was revealed that PCA analysis was more effective at distinguishing between the two groups of oils than analyzing each parameter separately. Thus, analyzing all results together by PCA (DSC, composition, and chemical oxidative stability), it was possible to differentiate both groups of oils in terms of seeds freshness.

## Figures and Tables

**Figure 1 foods-12-00135-f001:**
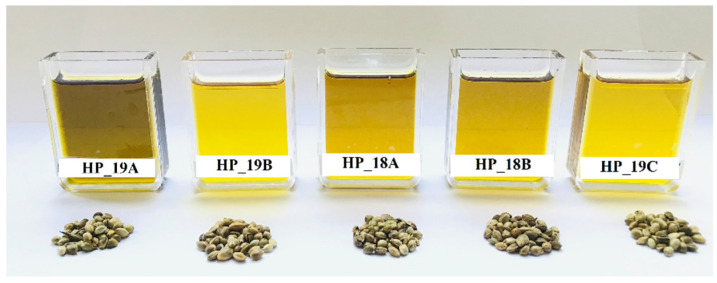
Hemp seeds and cold-pressed hemp seed oils. Group 1 oils (HL_18A, HL_18B) and group 2 oils (HL_19A, HL_19B, HL_19C).

**Figure 2 foods-12-00135-f002:**
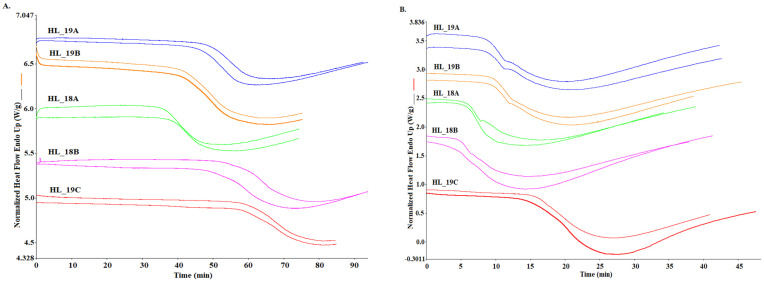
DSC curves obtained for isothermal analysis at (**A**) 120 °C and (**B**) 140 °C temperature program for cold-pressed hemp seed oil. Group 1 oils (HL_18A, HL_18B) and group 2 oils (HL_19A, HL_19B, HL_19C).

**Figure 3 foods-12-00135-f003:**
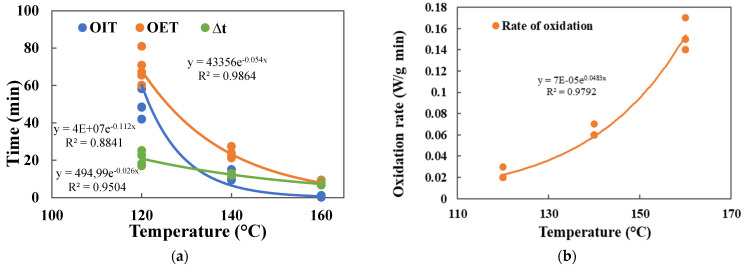
Relationship between temperatures and parameters of DSC isothermal measurement for 120, 140, and 160 °C: (**a**) for OIT, OET, and oxidation duration ∆t; (**b**) for rate of oxidation.

**Figure 4 foods-12-00135-f004:**
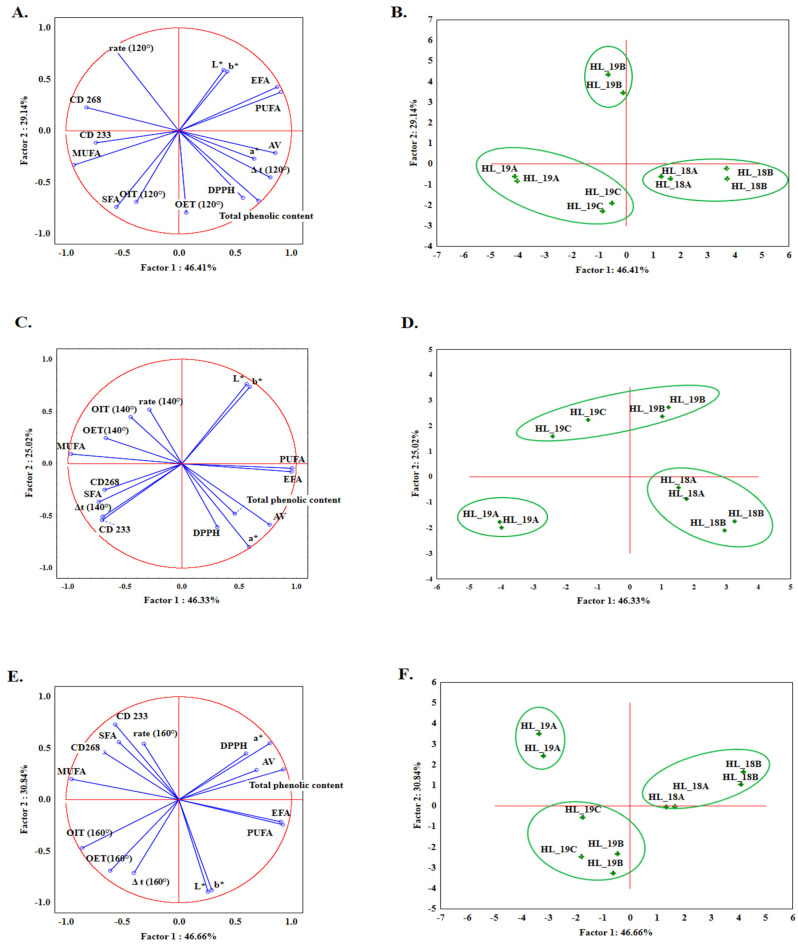
PCA analysis, the loading plots for PC1 and PC2 with projection of the variables: DSC parameters determined at 120 °C (**A**), 140 °C (**C**), and 160 °C (**E**). and projection of cases showing the distribution of hemp seed oil samples analyzed at 120 °C (**B**), 140 °C (**D**), 160 °C (**F**).

**Table 1 foods-12-00135-t001:** Color determination using a spectrophotometer.

CIE LAB L*, a*, b*	*Henola* Cultivar Oil Samples				
	Group 1		Group 2		
	HL_18A	HL_18B	HL_19A	HL_19B	HL_19C
L*	47.92 ^d^ ± 0.01	41.11 ^b^ ± 0.01	33.00 ^a^ ± 0.02	53.46 ^e^ ± 0.01	44.94 ^c^ ± 0.01
a*	10.06 ^d^ ± 0.01	11.81 ^e^ ± 0.01	7.73 ^c^ ± 0.01	5.82 ^b^ ± 0.01	5.68 ^a^ ± 0.01
b*	80.77 ^d^ ± 0.09	69.77 ^b^ ± 0.04	56.48 ^a^ ± 0.04	88.20 ^e^ ± 0.08	74.99 ^c^ ± 0.06

^abcde^—different superscript letters indicate significant differences (*p* ≤ 0.05) within rows. Color measurement using L*a*b* CIELAB color space. L* (lightness or darkness) ranges from black to white (0–100); a* demonstrates color red (a* > 0) or green (a* < 0); b* demonstrated color yellow (b* > 0) or blue (b* < 0). Group 1 oils (HL_18A, HL_18B) and group 2 oils (HL_19A, HL_19B, HL_19C).

**Table 2 foods-12-00135-t002:** Fatty acid composition expressed as a percentage of total fatty acid (%) of cold-pressed hemp seed oils of the *Henola* cultivar.

Fatty Acids	*Henola* Cultivar Oil Samples				
	Group 1		Group 2		
	HL_18A	HL_18B	HL_19A	HL_19B	HL_19C
∑SFA	10.91 ^b^ ± 0.03	10.65 ^a^ ± 0.06	11.03 ^b^ ± 0.01	10.56 ^a^ ± 0.06	10.92 ^b^ ± 0.12
∑MUFA	13.4 ^b^ ± 0.00	12.58 ^a^ ± 0.00	14.59 ^d^ ± 0.00	13.33 ^b^ ± 0.02	14.18 ^c^ ± 0.06
∑PUFA	75.45 ^c^ ± 0.02	76.57 ^d^ ± 0.06	74.14 ^a^ ± 0.00	75.76 ^c^ ± 0.01	74.67 ^b^ ± 0.18
∑EFA	73.43 ^b^ ± 0.04	74.33 ^c^ ± 0.06	72.43 ^a^ ± 0.01	73.72 ^b^ ± 0.04	72.68 ^a^ ± 0.19
UFA/SFA	8.143	8.371	8.048	8.440	8.314
n-6/n-3	3.094	2.917	1.896	2.799	3.097

^abcd^—values are mean ± standard deviations of two measurements (*n* = 2); different superscript letters indicate significant differences (*p* ≤ 0.05) within rows. ∑SFA—total of saturated fatty acid. ∑MUFA—total of monounsaturated fatty acid. ∑PUFA—total of polyunsaturated fatty acid. ∑EFA—total of essential fatty acid. UFA/SFA—ratio of total of unsaturated fatty acid to total of saturated fatty acid. Group 1 oils (HL_18A, HL_18B) and group 2 oils (HL_19A, HL_19B, HL_19C).

**Table 3 foods-12-00135-t003:** Tocochromanols contents of cold-pressed hemp seed oils and antioxidant activity determined by measuring DPPH radical scavenging activity (% inhibition).

Tocochromanols and DPPH Scavenging Activity (%)	*Henola* Cultivar Oil Samples				
	Group 1		Group 2		
	HL_18A	HL_18B	HL_19A	HL_19B	HL_19C
α-T (mg/100 g)	5.38 ^b^ ± 0.16	6.19 ^c^ ± 0.06	5.35 ^b^ ± 0.03	4.36 ^a^ ± 0.11	5.37 ^b^ ± 0.16
β-T (mg/100 g)	0.14 ^b^ ± 0.16	nd	0.15 ^b^ ± 0.01	0.13 ^ab^ ± 0.02	0.11 ^a^ ± 0.01
γ-T (mg/100 g)	73.43 ^d^ ± 0.18	73.78 ^d^ ± 0.2	65.79 ^b^ ± 0.01	63.47 ^a^ ± 0.22	70.49 ^c^ ± 0.25
δ-T (mg/100 g)	1.14 ^a^ ± 0.18	1.47 ^d^ ± 0.03	1.23 ^bc^ ± 0.03	1.16 ^ab^ ± 0.05	1.31 ^c^ ± 0.01
TPC (mg/100 g)	80.08 ^d^ ± 0.36	81.43 ^e^ ± 0.16	72.51 ^b^ ± 0.04	69.11 ^a^ ± 0.39	77.27 ^c^ ± 0.35
PC-8 (mg/100 g)	0.26 ^a^ ± 0.04	nd	1.64 ^c^ ± 0.01	0.34 ^b^ ± 0.02	0.34 ^b^ ± 0.01
DPPH (µmol TE/g oil)	8.53 ^bc^ ± 0.6	9.11 ^c^ ± 0.43	7.83 ^ab^ ± 0.34	6.96 ^a^ ± 0.1	8.33 ^bc^ ± 0.41

^abcde^—values are mean ± standard deviations of three measurements (*n* = 3), different superscript letters indicate significant differences (*p* ≤ 0.05) within rows. TPC—Total phenolic content. PC-8— Plastochromanol-8. nd—not detected.

**Table 4 foods-12-00135-t004:** Acid values, peroxide values, p-anisidine values, TOTOX values, dienoic and trienoic acid values, and specific extinctions coefficient at UV wavelengths 233 and 268 for cold-pressed hemp seed oils.

Chemical Analysis	*Henola* Cultivar Oil Samples				
	Group 1		Group 2		
	HL_18A	HL_18B	HL_19A	HL_19B	HL_19C
AV (mg KOH/g)	18.53 ^b^ ± 0.46	21.16 ^c^ ± 0.47	10.31 ^a^ ± 0.07	10.30 ^a^ ± 0.03	10.33 ^a^ ± 0.06
PV (meq O_2_/kg)	17.53 ^d^ ± 0.12	13.79 ^b^ ± 0.04	18.80 ^e^ ± 0.13	15.81 ^c^ ± 0.20	11.01 ^a^ ± 0.37
p-AV	1.23 ^a^ ± 0.53	0.97 ^a^ ± 0.34	1.10 ^a^ ± 0.24	1.16 ^a^ ± 0.80	1.34 ^a^ ± 0.15
TOTOX	36.29 ^d^ ± 0.31	28.55 ^b^ ± 0.40	38.70 ^e^ ± 0.50	32.93 ^c^ ± 1.21	23.36 ^a^ ± 0.79
Dienoic acid value	0.11 ^a^ ± 0.01	0.11 ^a^ ± 0.01	0.22 ^b^ ± 0.00	0.11 ^a^ ± 0.00	0.10 ^a^ ± 0.00
Trienoic acid value	0.01 ^a^ ± 0.00	0.00 ^a^ ± 0.01	0.16 ^c^ ± 0.00	0.07 ^b^ ± 0.00	0.00 ^a^ ± 0.00
K_233_	0.16 ^a^ ± 0.01	0.16 ^a^ ± 0.01	0.30 ^b^ ± 0.00	0.16 ^a^ ± 0.00	0.15 ^a^ ± 0.00
K_268_	0.07 ^a^ ± 0.00	0.03 ^a^ ± 0.01	0.07 ^b^ ± 0.01	0.04 ^a^ ± 0.00	0.03 ^a^ ± 0.00

^abcde^—different superscript letters indicate significant differences (*p* ≤ 0.05) within rows. PV—peroxide value; p-AV—p-anisidine value; TOTOX—total oxidation indictor; K_233_ and K_268_—specific extinction coefficient at wavelengths λ = 233 and 268. Group 1 oils (HL_18A, HL_18B) and group 2 oils (HL_19A, HL_19B, HL_19C).

**Table 5 foods-12-00135-t005:** Isothermal assessment of 120, 140, and 160 °C temperature programs using differential scanning calorimetry for cold-pressed hemp seed oils.

DSC Isothermal Parameters		*Henola* Cultivar Oil Samples				
		Group 1		Group 2		
		HL_18A	HL_18B	HL_19A	HL_19B	HL_19C
Isothermal 120 °C	OIT (min)	48.22 ^b^ ± 1.30	41.89 ^a^ ± 0.90	48.49 ^b^ ± 0.50	41.95 ^a^ ± 0.00	58.18 ^c^ ± 0.63
	OET (min)	70.76 ^b^ ± 1.40	67.05 ^ab^ ± 1.85	65.43 ^ab^ ± 0.06	60.13 ^a^ ± 4.04	80.88 ^c^ ± 1.00
	Δt (min)	22.53 ^ab^ ± 0.10	25.17 ^b^ ± 0.95	16.94 ^a^ ± 0.56	18.19 ^ab^ ± 4.05	22.70 ^ab^ ± 0.37
	Rate of oxidation (W/g min)	0.02 ^ab^ ± 0.00	0.02 ^a^ ± 0.00	0.02 ^bc^ ± 0.00	0.03 ^c^ ± 0.00	0.02 ^a^ ± 0.00
Isothermal 140 °C	OIT (min)	9.33 ^a^ ± 0.06	9.44 ^a^ ± 0.56	10.15 ^a^ ± 0.05	9.39 ^a^ ± 0.23	14.99 ^b^ ± 1.05
	OET (min)	21.36 ^a^ ± 0.06	21.74 ^a^ ± 0.58	23.85 ^b^ ± 0.03	21.17 ^a^ ± 0.4	27.4 ^c^ ± 0.34
	Δt (min)	12.03 ^a^ ± 0.00	12.30 ^a^ ± 0.01	12.91 ^a^ ± 0.08	11.78 ^a^ ± 0.62	12.41 ^a^ ± 1.39
	Rate of oxidation (W/g min)	0.06 ^a^ ± 0.00	0.06 ^a^ ± 0.00	0.06 ^a^ ± 0.00	0.05 ^ab^ ± 0.02	0.07 ^b^ ± 0.01
Isothermal 160 °C	OIT (min)	0.6 ^b^ ± 0.06	0.08 ^a^ ± 0.01	0.84 ^c^ ± 0.01	0.98 ^cd^ ± 0.05	1.13 ^d^ ± 0.05
	OET (min)	7.68 ^ab^ ± 0.09	6.71 ^a^ ± 0.13	7.76 ^ab^ ± 0.55	8.56 ^ab^ ± 0.63	9.38 ^b^ ± 0.99
	Δt (min)	7.07 ^a^ ± 0.16	6.62 ^a^ ± 0.14	6.92 ^a^ ± 0.54	7.57 ^a^ ± 0.68	8.26 ^a^ ± 1.05
	Rate of oxidation (W/g min)	0.14 ^a^ ± 0.00	0.15 ^a^ ± 0.00	0.17 ^a^ ± 0.04	0.15 ^a^ ± 0.02	0.14 ^a^ ± 0.03

^abcd^—different superscript letters within rows indicate significant differences (*p* ≤ 0.05). OIT—Oxidation induction time, OET—oxidation end time, Δt—Length of oxidation. Group 1 oils (HL_18A, HL_18B) and group 2 oils (HL_19A, HL_19B, HL_19C).

**Table 6 foods-12-00135-t006:** Non-isothermal assessment of heating rate 2 and 5 °C/min temperature programs using differential scanning calorimetry for cold-pressed hemp seed oils.

DSC Non-Isothermal Parameters		*Henola* Cultivar Oil Samples				
		Group 1		Group 2		
		HL_18A	HL_18B	HL_19A	HL_19B	HL_19C
(2 °C /min)	T_on_	147.54 ^ab^ ± 0.42	146.27 ^ab^ ± 0.23	146.61 ^ab^ ± 0.22	145.26 ^a^ ± 0.52	148.14 ^b^ ± 1.33
	T_end_	163.29 ^a^ ± 1.3	160.26 ^a^ ± 0.8	154.76 ^a^ ± 5.75	158.61 ^a^ ± 1.21	162.02 ^a^ ± 0.00
(5 °C /min)	T_on_	160.39 ^c^ ± 0.21	158.94 ^a^ ± 0.3	159.96 ^bc^ ± 0.33	159.34 ^ab^ ± 0.2	163.44 ^d^ ± 0.16
	T_end_	183.25 ^a^ ± 0.9	184.9 ^a^ ± 1.74	183.2 ^a^ ± 1.17	184.45 ^a^ ± 0.24	188.14 ^a^ ± 1.57

^abcd^—different superscript letters within rows indicate significant differences (*p* ≤ 0.05). Oxidative stability parameters: Onset of temperature (T_on_) and end temperature (T_end_) determination at 2 and 5 °C/min non-isothermal condition. Group 1 oils (HL_18A, HL_18B) and group 2 oils (HL_19A, HL_19B, HL_19C).

**Table 7 foods-12-00135-t007:** Pearson’s correlation coefficients between DSC oxidation parameters and MUFA, PUFA, AV, and a* values.

DSC Parameters	MUFA	PUFA	AV	a*
OIT (120 °C)	0.677	−0.687	−0.428 *	−0.461 *
OET (120 °C)	0.296 *	−0.319 *	−0.031 *	−0.128 *
OIT (140 °C)	0.492 *	−0.468 *	−0.454 *	−0.545 *
OET (140 °C)	0.663	−0.648	−0.514 *	−0.519 *
OIT (160 °C)	0.740	−0.712	−0.921	−0.959
OET (160 °C)	0.506 *	−0.458 *	−0.724	−0.837
T_on_ (2 °C/min)	0.352 *	−0.376 *	0.022 *	−0.016 *
T_on_ (5 °C/min)	0.552 *	−0.555 *	−0.423 *	−0.532 *

OIT—oxidation induction time (min), OET—oxidation end time (min) for various isothermal temperatures: 120, 140, and 160 °C; MUFA—monounsaturated fatty acids, PUFA—polyunsaturated fatty acids, AV—acid values (mg KOH/g), a*—color parameter, T_on_—onset of temperature for non-isothermal condition (2 and 5 °C/min), * Correlation not significant statistically, (α = 0.05).

## Data Availability

The data presented in this study are available upon reasonable request.

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
