# Peer review of "Assessment of Hemp Seed Oil Quality Pressed from Fresh and Stored Seeds of Henola Cultivar Using Differential Scanning Calorimetry"

_foods, 2022, doi:10.3390/foods12010135_

Round 1
Reviewer 1 Report
The manuscript entitled “Assessment of hemp seed oil quality pressed from fresh and stored seeds 2 of Henola cultivar using differential scanning calorimetry” reports an interesting evaluation of the hemp seed oils quality from seeds from different years of harvesting. The interest for the readers could be very high; however, in my opinion, the authors have to address some critical points before recommending the publication of this manuscript. In particular, the main point is related to the storage of the seeds. The authors declare that the oils were from “stored” and “fresh” hemp seeds, but the harvesting campaigns were 2018 and 2019 while the cold-pressing was performed in 2020. In this regard, I have two points: Firstly, why the authors considered the 2019 seeds as “fresh”? I supposed they had been stored in some way until February 2020; the second point is related to the storage conditions of the seeds from 2018 and of the seeds from 2019: are they the same (temperature, relative humidity, etc.)? It is necessary to report these conditions in order to interpret and compare the results.
Materials and methods:
Paragraph 2.2.1: Please, expressed the weight of the oil as g and not as “drop”, since it is not a unit of measurement.
Line 115: Report the fatty acids in the mixture with the relative CAS number, as well as the State of the company.
Line 167: Please, write 232 and 268 as subscript, check it during the manuscript.
Line 171: A space is missing before the parenthesis and after the word "radicals".
Results:
Lines 213-214: The phrase "since .." seems to be interrupted, I suggest the authors to rephrase it.
Fatty acids: Since the authors used a standard mix for the identification, I encourage they to submit, e.g. as a supplementary file, a table which reports each fatty acid in the oils. It could be interesting for the readers, and also to improve the discussion of the results.
Line 297: Please, wrote the “2” of “O2” as subscript. Check it during all the manuscript.
Discussion: The discussion must be implemented and improve. There are some data that are presented but not discussed in depth, for example PV and K. I think the discussion can be more in-depth, even with a greater comparison with the literature, in this sense some recent studies have not been included among the references list, while some others (especially if used only once and in combination with others) can be avoided.
Author Response
Dear Reviewer,
thank you very much for taking the time and effort to help us improve the quality of the paper. We appreciate all insightful comments and suggestions of the Reviewer, which we have carefully considered and corrected. Our response is listed below:
The manuscript entitled “Assessment of hemp seed oil quality pressed from fresh and stored seeds 2 of Henola cultivar using differential scanning calorimetry” reports an interesting evaluation of the hemp seed oils quality from seeds from different years of harvesting. The interest for the readers could be very high; however, in my opinion, the authors have to address some critical points before recommending the publication of this manuscript. In particular, the main point is related to the storage of the seeds. The authors declare that the oils were from “stored” and “fresh” hemp seeds, but the harvesting campaigns were 2018 and 2019 while the cold-pressing was performed in 2020. In this regard, I have two points: Firstly, why the authors considered the 2019 seeds as “fresh”? I supposed they had been stored in some way until February 2020; the second point is related to the storage conditions of the seeds from 2018 and of the seeds from 2019: are they the same (temperature, relative humidity, etc.)? It is necessary to report these conditions in order to interpret and compare the results.
Reply: The seeds were harvested at the end of November (November, 29th) and oils were pressed at the beginning of February (February, 3rd), so there were only two months of storage. All seeds were stored at the same conditions of temperature and humidity. In industrial practice, this is a standard procedure, the oils are pressed from the last harvest until the next harvest of seeds, so after two months of storage, we considered that these seeds are fresh.
Materials and methods:
Paragraph 2.2.1: Please, expressed the weight of the oil as g and not as “drop”, since it is not a unit of measurement.
Reply: It has been replaced “drop” by “around 15 mg”
Line 115: Report the fatty acids in the mixture with the relative CAS number, as well as the State of the company.
Reply: It has been added: “Supelco 37 Component FAME Mix, CRM47885, Sigma-Aldrich, St. Louis, MO, USA)
Line 167: Please, write 232 and 268 as subscript, check it during the manuscript.
Reply: It is corrected throughout whole manuscript
Line 171: A space is missing before the parenthesis and after the word "radicals".
Reply: It is corrected
Results:
Lines 213-214: The phrase "since .." seems to be interrupted, I suggest the authors to rephrase it.
Fatty acids: Since the authors used a standard mix for the identification, I encourage they to submit, e.g. as a supplementary file, a table which reports each fatty acid in the oils. It could be interesting for the readers, and also to improve the discussion of the results.
Reply: The table with each fatty acid content is attached as a supplementary file. (Table S1)
Line 297: Please, wrote the “2” of “O2” as subscript. Check it during all the manuscript.
Reply: It is corrected
Discussion: The discussion must be implemented and improve. There are some data that are presented but not discussed in depth, for example PV and K. I think the discussion can be more in-depth, even with a greater comparison with the literature, in this sense some recent studies have not been included among the references list, while some others (especially if used only once and in combination with others) can be avoided.
Reply: The discussion has been improved by adding the text:
“Data presented in Table 4 shows the samples stability determined using different conventional chemical methods. Peroxide values represents the presence of peroxides and hydroperoxides (known as primary oxidation products), whence p anisidine values shows the concentration of unsaturated aldehydes (detection at 350 nm absorption), which can express the secondary oxidation metabolites present in samples. Considering the primary oxidation condition, samples shows significant differences (p ≤ 0.05) regardless the freshness of the seeds, whence for secondary oxidation phenomena, the characteristics are opposite, thus pronounce the overall freshness of the oil samples. Hence, oil sample HP_19C showed highest stability simultaneously throughout all chemical measurement, which in turn has been supported by the results obtained from isothermal (OIT) and dynamic (Ton) parameters from DSC. Different approach presented by authors shows the simultaneous increase in primary and secondary oxidation products after roasting them in different temperatures [7] or during accelerated oxidation [18]. Alternative chemical analysis to determine oxidative stability paved the results of conjugated dienoic and trienoic acid (%) values, and the ultraviolet absorption measured at wavelengths λ = 233 (K233) and λ =268 (K268) presented for oil samples shows the primary and secondary oxidation products for both groups of oils. Although, these data shows the notifiable presence of primary deterioration for all samples, however they did not show any significant differences (p > 0.05) between the seeds quality of stored and fresh pressed oils.”
Reviewer 2 Report
1. please include more scientific evidences to support your review in section 1.0
2. Please provide graphical explanation to summarize your methodology with the original images of the samples and etc
3. It would be better to provide the image of the samples for color measurement testing
4. Please include the chemical equation to support your results.
Author Response
Dear Reviewer,
thank you very much for taking the time and effort to help us improve the quality of the manuscript. We appreciate all insightful comments and suggestions of the Reviewer, which we have carefully considered and corrected. Our response is listed below:
- please include more scientific evidences to support your review in section 1.0
Reply: In the introduction part the text was added “
“Apart from pressing them for oil, hemp seed foods are popular throughout the Baltic states, North America and Europe, middle East and China. As a result, storing the seeds for various purpose throughout seasons is a common phenomenon [7]. However, the unique composition of hemp seeds and oils pressed from them has been highlighted in literatures with the abundance of chlorophyll and PUFA content, which results rapid complex series of chemical reactions leads to deterioration of the seeds and oils [18]. Study shows, thermal characteristics of hemp seed oils analyzed using Thermogravimetric analysis (TGA) [19], or Pressure Differential Scanning Calorimeter (PDSC) [20] are able to provide copper-bottomed database to assess the oxidative stability parameters in isothermal and non-isothermal conditions. Additionally, the comprehensive analysis of such vegetable oils using DSC has been provided with significant results to characterize oils quality in terms of stability [12]”
Please provide graphical explanation to summarize your methodology with the original images of the samples and etc
Reply: The graphical abstract has been prepared.
- It would be better to provide the image of the samples for color measurement testing
Reply: The image of all hempseed oil samples has been added as the figure 1
- Please include the chemical equation to support your results.
Reply: The chemical equations referring to the parameters of OIT and OET were inserted in the text of manuscript.
“Oxidation induction time (OIT) indicates the initiation of oxidation, which stage is the reaction between the radicals, formed during the induction time and PUFA as shown in equation (2). A decrease in the heat flow is related to the propagation stage, where oxygen is consumed in the reaction according to equations (3). Oxidation end time (OET) expresses the termination stage, where stable products are formed (equations 4).
Initiation R-H + RË™→ RË™ + HË™ (2)
Propagation RË™ + O2 → ROOË™
ROOË™ + RH → ROOH + RË™ (3)
Termination (stable products) ROOË™ + ROOË™ → ROOR + O2
ROË™ + RË™ → ROR
RË™ + RË™ → RR (4)”